# Pathological changes in oral epithelium and the expression of SARS-CoV-2 entry receptors, ACE2 and furin

**Osnat Grinstein-Koren**[1], **Michal Lusthaus**[1], **Hilla Tabibian-Keissar**[2], **Ilana Kaplan**[1,3], **Amos Buchner**[1], **Ron Ilatov**[4], **Marilena Vered**●[1,2]*, **Ayelet Zlotogorski-Hurvitz**[1,5]

**1** Department of Oral Pathology, Oral Medicine and Maxillofacial Imaging, Goldschleger School of Dental Medicine, Faculty of Medicine, Tel-Aviv University, Tel-Aviv, Israel, **2** Institute of Pathology, The Chaim Sheba Medical Center, Tel Hashomer, Ramat Gan, Israel, **3** Institute of Pathology, Rabin Medical Center, Petach-Tikva, Israel, **4** Goldschleger School of Dental Medicine, Tel-Aviv University, Tel-Aviv, Israel, **5** Department of Oral and Maxillofacial Surgery, Rabin Medical Center, Petach-Tikva, Israel

* mvered@tauex.tau.ac.il

**Data Availability Statement:** All relevant data are within the manuscript and its Supporting Information files.

## Abstract

### Background

Expression of angiotensin-converting enzyme (ACE)-2 and co-factors like furin, play key-roles in entry of SARS-CoV-2 into host cells. Furin is also involved in oral carcinogenesis. We investigated their expression in oral pre-malignant/malignant epithelial pathologies to evaluate whether ACE2 and furin expression might increase susceptibility of patients with these lesions for SARS-CoV-2 infection.

### Methods

Study included normal oral mucosa (N = 14), epithelial hyperplasia-mild dysplasia (N = 27), moderate-to-severe dysplasia (N = 24), squamous cell carcinoma (SCC, N = 34) and oral lichen planus (N = 51). Evaluation of ACE2/furin membranous/membranous-cytoplasmic immunohistochemical expression was divided by epithelial thirds (basal/middle/upper), on a 5-tier scale (0, 1—weak, 1.5 –weak-to-moderate, 2—moderate, 3—strong). Total score per case was the sum of all epithelial thirds, and the mean staining score per group was calculated. Real time-polymerase chain reaction was performed for ACE2-RNA. Statistical differences were analyzed by One-way ANOVA, significance at p<0.05.

### Results

All oral mucosa samples were negative for ACE2 immuno-expression and its transcripts. Overall, furin expression was weakly present with total mean expression being higher in moderate-to-severe dysplasia and hyperplasia-mild dysplasia than in normal epithelium (p = 0.01, each) and SCC (p = 0.008, p = 0.009, respectively).

### Conclusions

Oral mucosa, normal or with epithelial pathologies lacked ACE2 expression. Furin was weak and mainly expressed in dysplastic lesions. Thus, patients with epithelial pathologies

**Funding:** The study was supported by the Herb and Ed Stein Chair in Oral Pathology, Tel Aviv University. The funders had no role in study design, data collection and analysis, decision to publish, or preparation of the manuscript.

do not seem to be at higher risk for SARS-CoV-2 infection. Overall, results show that oral mucosae do not seem to be a major site of SARS-CoV-2 entry and these were discussed vis-à-vis a comprehensive analysis of the literature.

## Introduction

The severe acute respiratory syndrome coronavirus 2 (SARS-CoV-2), which was first reported in December 2019 in Wuhan, China, was found to be responsible for the coronavirus disease 2019 (COVID-19) that became a worldwide pandemic [1]. This is an enveloped single stranded RNA virus with a complete genome of about 30,000 nucleotides, that translates both structural and non-structural proteins [2]. Viral structural proteins include the spike (S), envelope (E), membrane (M) and nucleocapsid (N) types [3]. The first step in SARS-CoV-2 infection occurs when the surface spike protein (the S1 part), through the receptor binding domain, attaches to the host membranal angiotensin-converting enzyme 2 (ACE2), which thus confers it a major role as the host receptor for viral attachment and transmission [4, 5]. This is followed by cleavage between the S1 and S2 protein complex by host protease furin, consequently exposing the fusion peptide of the S2 protein, which then undergoes cleavage by transmembrane serine protease 2 (TMPRSS2), now allowing attachment of the viral particle to the host cell plasma membrane. This results in structural alteration of the S2 protein, which permits fusion of the viral envelope with the host cell plasma membrane, finally followed by the release of viral genome into the host cytoplasm [4–7]. Therefore, a successful entry of the virus into the host cells requires that all three host molecular factors (i.e., ACE2 as the major receptor, and furin and TMPRSS2 as indispensable auxiliary host cell-derived cleavage proteases) to be expressed simultaneously on the cell membranes, with an extracellular domain accessible to the viral attaching proteins.

The concept regarding the status of the oral cavity in the pathogenesis of COVID-19 has evolved from of the oral cavity having a passive role in the transmission of SARS-CoV-2 to it being an independent source of virus transmissibility within and among subjects [8–11]. This was based on reports that oral lining epithelial cells express the SARS-CoV-2 entry receptor–ACE2 and its co-factors–furin and TMPRSS2, especially the lining epithelium of the tongue, as well as the parenchyma of the minor salivary glands [12–16].

In addition to the functions performed by ACE2 and furin in the context of SARS-CoV-2 attachment to host cells, both proteins are known to play other roles in the human body, in both physiological and pathological conditions. Physiologically, ACE2 is known for its role as a major regulator of blood pressure homeostasis [17]. Among the pathological conditions, inflammatory reactions have been shown to upregulate expression of ACE2 [18]. Furin is a cellular endoprotease that activates proteolysis of a large numbers of proprotein substrates ranging between its function as a housekeeping protein to a crucial role in various disease processes, including bird flu, dementia, Ebola fever and cancer, among others [19]. Furin has been investigated also in regard to head and neck carcinogenesis [20, 21]. It has been found to be extensively involved in tumor progression in tongue squamous cell carcinoma (SCC), as its expression together with that of vascular endothelial growth factor C (VEGF-C) were considerably higher in pre-invasive lesions and invasive neoplasia. Furin expression was also found to be correlated with increased micro-vessel density in precursor lesions and SCC of the tongue. In vitro, it has been shown that furin had the potential to convert SCC-related matrix metallopeptidase 14 (MMP14) zymogen into its active form, required for collagen degradation by the tumor cells [21]. In view of furin playing important cellular roles in both oral

carcinogenesis and SARS-CoV-2 attachment, we hypothesized that its expression in the context of oral epithelial pre-malignant lesions and oral SCC (OSCC), might increase the risk of patients with these entities for SARS-CoV-2 infection. Furthermore, based on the finding that inflammatory conditions upregulate expression of ACE2 [18], we raised the possibility that also patients diagnosed with chronic oral inflammatory/immune conditions, like oral lichen planus (OLP), might be more susceptible to infection by SARS-CoV-2. Therefore, we aimed to investigate expression of ACE2 and furin in these types of oral lesions and to discuss findings in relation to those reported in the literature and in the context of the role of the oral cavity as a port of entry for the SARS-CoV-2.

## Material and methods

Study was approved by the Ethics Committee of Tel Aviv University #1698, 24th June 2020. Study design was performed according to STROBE reporting standards for observational studies. By routine, all patients referred to the dental clinics of the School of Dentistry at Tel Aviv University, sign on an informed consent form during the first admission and anamnesis procedure, whereby they approve the use of their medical records (clinical, radiological and microscopic) for teaching and research purposes in an anonymized manner. Minors were not included in the study. The archive files of the laboratory of oral pathology were accessed between July 1st, 2020 and October 30th, 2020. Cases were selected per key-words of diagnostic entities (please see below), their original numbers and those of the corresponding paraffin blocks were coded randomly with running numbers starting from 1, and allocated to each of the study groups. The slides prepared from these blocks for immunohistochemical stains were also labelled by the coded numbers, so no further connection could be made to the original block numbers when the slides were analyzed. The original list of the coded cases was permanently deleted once the slides were coded, eliminating any possibility of identification of individual patients during or after data collection.

Retrieved were cases diagnosed as epithelial hyperplasia consisting of an increased width of the spinous layer with normal cellular maturation lacking dysplastic changes; epithelial hyperplasia with dysplasia was assessed as mild when changes involved only the basal third of the epithelial width, moderate–basal and middle thirds, and severe–dysplastic changes involved the upper third of the epithelial width; SCC of the tongue and OLP; samples of normal oral epithelium were obtained from biopsies of amalgam tattoo. Due to the occasional presence of a gradual escalation in epithelial changes in adjacent areas even in the same sample, diagnostic entities were grouped as following: epithelial hyperplasia-to-mild dysplasia and moderate-to-severe dysplasia. As the SARS-CoV-2 is primarily a virus of the respiratory tract [1–3], we included samples of upper respiratory mucosa (nose and maxillary sinus), collected from cases of inflammatory conditions, which served as controls. All cases comprised of tissues that were referred to the oral pathology laboratory before the COVID-19 pandemic. They were routinely fixed in 10% buffered formalin (24–48 h) and embedded in paraffin. For the immunohistochemical identification of ACE-2 and furin, a monoclonal antibody SN0754 (Novus Biologicals, Centennial, CO, USA; catalog #NBP2-67692, 1:200) and a polyclonal antibody (ProteinTech, Manchester, UK; catalog #18413-1-AP, 1:200), respectively, were used. Normal kidney tissue served as positive control for ACE2 and salivary gland parenchyma as positive control for furin. Staining results were validated in comparison to the Human Protein Atlas (HPA), version 22.0 that has been issued in December 2022 (https://www.proteinatlas.org) (Fig 1).

Histomorphometric analysis for the immunohistochemical expression of ACE2 and furin focused on the oral lining epithelium, where its width was imaginary divided into equal thirds

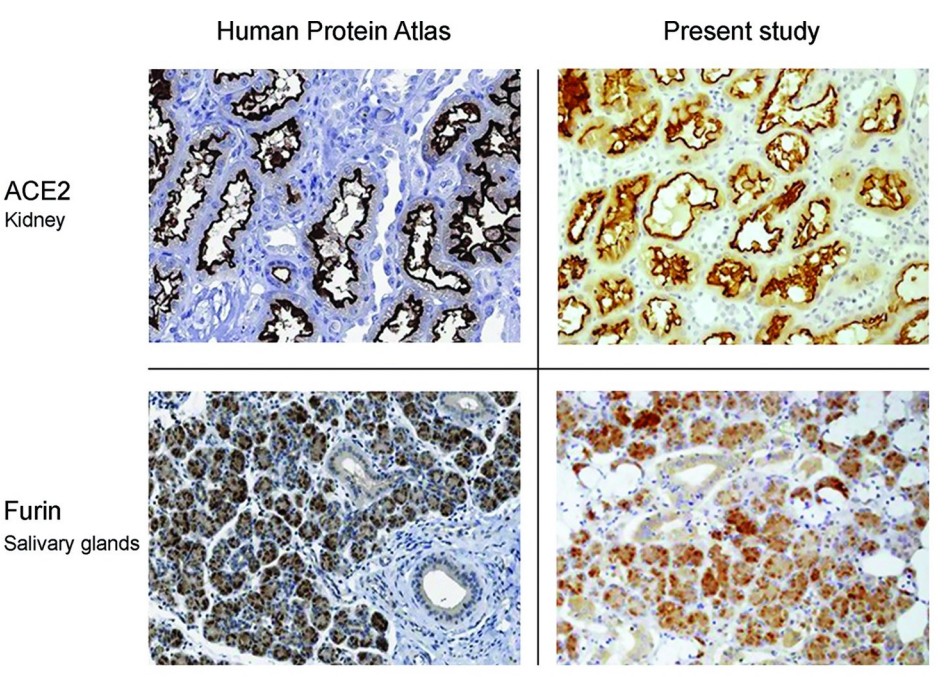

**Fig 1. Control tissues for ACE2 and furin.** Standard control immunohistochemical stain for ACE2 was performed on kidney tissue. In both the section from the HPA (upper left) and our section (upper right), strong ACE2 stain (reflected as a brown-colored reaction) is seen at the luminal surface of the lining cells of the tubules. For furin, salivary glands were used as the standard control. In both the section from the HPA (lower left) and our section, salivary gland acinar cells were found to be positively stained.

—basal, middle and upper (excluding keratin), similar to the system used for grading of dysplasia. Intensity of staining per each third was assessed on a 5-tier scale as follows: 0 –no staining, 1 –weak, 1.5 –weak-to-moderate, 2 –moderate, and 3 –strong. The total score per case was the sum of scores for all thirds (basal + middle + upper). For cases of SCC, a modification of the "thirds" system was performed and these were evaluated along the axis of invasion of the tumor: the entire distance, starting from below the level of the adjacent oral lining epithelium to the deepest point/front of the invading tumor, was imaginary divided into three equal thirds: the upper third was that closest to the oral lining epithelium, followed by the middle and the deepest. For both investigated markers, only positive membranous and cytoplasmic immunohistochemical staining was recorded; positive nuclear staining was considered non-specific. All slides were examined by three authors (MV, AZH, IK), with scores presenting joint accord among them.

Gene expression of *ACE2* was measured by real-time polymerase chain reaction (RT-PCR). Paraffin blocks that were submitted for the PCR study were first checked for the presence of a satisfactory amount of tissue. Total RNA was extracted from formalin-fixed, paraffin-embedded (FFPE) tissue samples following the protocol described by Tian et al., [22] using the RNeasy FFPE kit (Qiagen, catalogue #73504; Hilden, Germany). From the tissue blocks, 5–15 sections (depending on tissue size) 3.5–4 μ width, were cut and deparaffinized by adding 150 μl of protein kinase D buffer (Qiagen) and then incubation at 90˚C for 3–4 min followed by 30 sec centrifugation at full speed. The expelled paraffin ring was removed. In order to increase tissue digestion and RNA recovery, we elongated incubation time of samples with proteinase K (50 μl/sample) to 2–4 h at 56˚C, followed by a 15 min incubation at 80˚C and cooling on ice for 2 min. The rest of the RNA extraction procedure was performed according

to the manufacturer protocol, 30–50 μl of PCR grade water was used for elusion, followed by incubation for several minutes at room temperature and centrifugation. RNA concentration was measured by NanoDrop[TM] 1000 spectrophotometer (Thermo Fisher Scientific, Waltham, MA, USA). One microgram of total RNA was used in all present samples for cDNA synthesis (0.25 mg RNA is the minimal required amount) using the Applied Biosystems kit (catalogue #4368814, Thermo Fisher Scientific Baltics, Vilnus, Lithuania). We used predesigned TaqMan Gene Expression Assay for ACE2, Hs00222343_m1 (Applied Biosystems, Foster City, CA, USA). Results were planned to be analyzed by the $2^{-\Delta\Delta Ct}$ method; normalization was done in relation to *glyceraldehyde-3-phosphate dehydrogenase* (*GAPDH*) gene expression, Hs99999905_m1 (Applied Biosystems). The resulting data were analyzed using StepOne Plus Real-Time PCR System (Applied Biosystems, Waltham, MA, USA). Mean value of $\leq 36$ cycle threshold (Ct) of fluorescent signals above background level, was considered as a positive signal [23].

## Statistical analysis

Differences in the mean scores of immunohistochemical expression of ACE2 and furin among the various study groups was analyzed by One-way ANOVA followed by Bonferroni correction, in order to overcome type I error (false positive) in the computation of multiple comparisons. Significance was set at $p < 0.05$. Analysis was carried out using the SPSS software, version 25 (IBM, Chicago, Il, USA).

## Results

The following study groups were included: epithelial hyperplasia-to-mild dysplasia (HP, N = 27), moderate-to-severe dysplasia (Mod-Sev, N = 24), SCC (N = 34), OLP (N = 51), normal oral epithelium (Normal, N = 14) and upper respiratory mucosa of the nose and maxillary sinus (N = 5).

### 1. Immunohistochemical stains

ACE2 –the oral epithelium of all samples analyzed with the various degrees of dysplasia and carcinoma, interface inflammation in OLP, and the normal controls, were all immuno-negative (Fig 2).

Only the upper respiratory tract epithelium from the maxillary sinus and nose showed expression of ACE2 in the area immediately underlying the cilia, which was consistent with the staining pattern shown in HPA (Fig 2). In areas where the ACE2-positive respiratory epithelium was hyperplastic and associated with inflammation, it either showed detached parts of cells or entire cells that shed into the lumen area. Only some of the small blood vessels within the parenchyma of the submandibular salivary glands were positive for ACE2 (Fig 3).

Furin expression was found in all epithelial types of lesions, including normal oral epithelium, in the cytoplasmic compartment. The mean total score of furin expression only of Mod-Sev and HP were slightly over the value of 1, implying an overall weak expression (Fig 4 and S1 File). Expression of furin was overall low in all thirds in SCC and in the Normal group it was absent from the middle and superficial thirds. The mean total score of furin in SCC was significantly lower than in Mod-Sev (p = 0.008) and HP (p = 0.009). The expression in Normal was significantly lower than in Mod-Sev and HP (p = 0.01). The basal third showed a higher expression compared to the middle and upper thirds in lesions of epithelial dysplasia (all severities) and in OLP. Differences in the mean basal score of furin expression were found between SCC and Mod-Sev (p = 0.032), SCC and HP (p = 0.027), between Normal and Mod-Sev (p = 0.043) and Normal and HP (p = 0.036). There were no significant differences in the mean scores of furin expression in the middle and upper thirds among types of lesions.

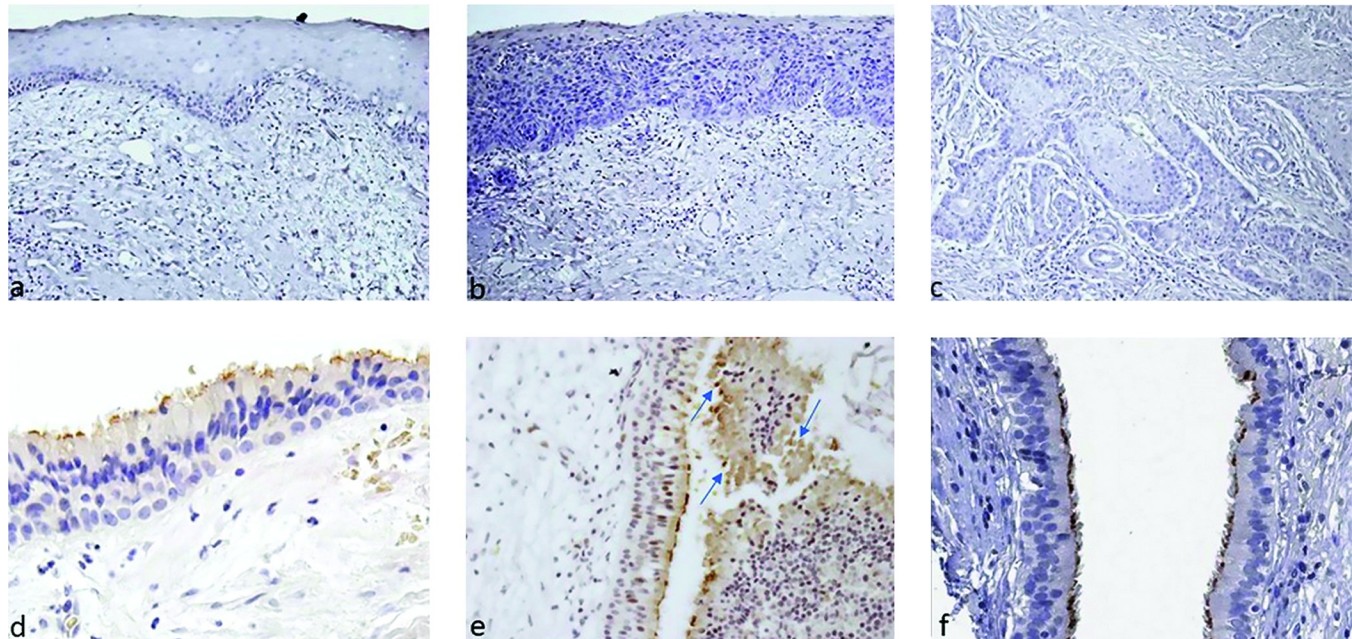

**Fig 2. ACE2 expression in oral epithelium.** Normal control (a), moderate-to-severe dysplasia (b), squamous cell carcinoma (c), upper respiratory mucosa (maxillary sinus) (d), inflamed upper respiratory mucosa (maxillary sinus) showing ACE2-positively stained detached parts of cells or whole cells (arrows) (e), upper respiratory mucosa Human Protein Atlas (f) (a-c, e—original magnification x200; d—original magnification x400).

Upper respiratory mucosa also showed positive expression of furin. Fig 5 shows furin immunostaining in representative samples of various lesions and control tissues.

## 2. RT-PCR study

Samples of normal oral mucosa, HP, Mod-Sev, SCC, OLP, nasal and maxillary mucosa and kidney were submitted for RT-PCR analysis. Positive signals were received only from the kidney tissue, all other samples were negative, therefore no further analysis of the results (including ΔCT, 2^-ΔΔCT, relative fold change) was performed.

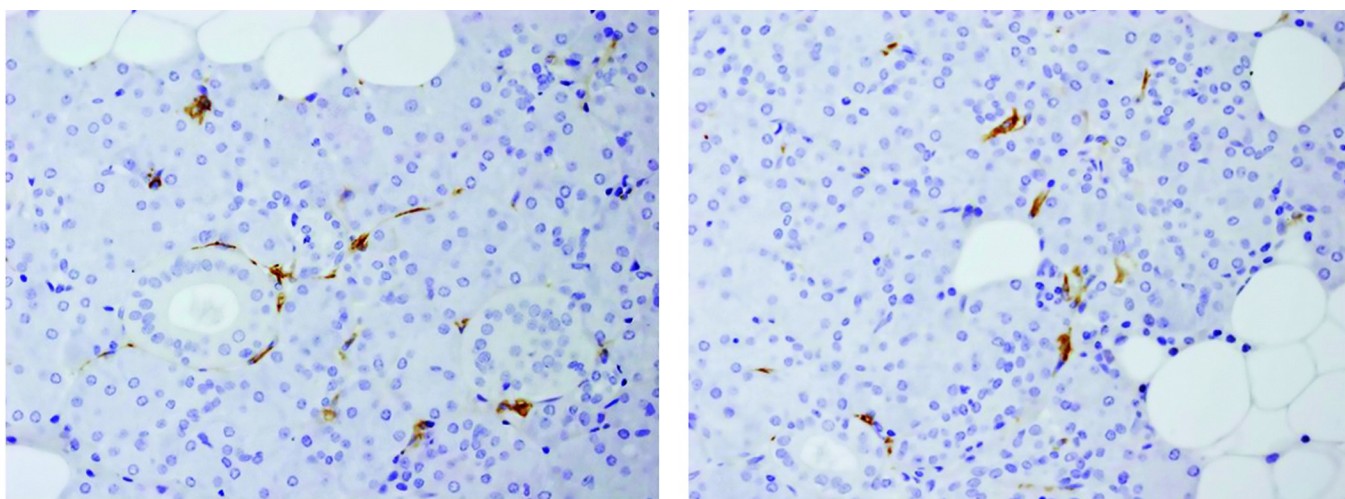

**Fig 3. Submandibular salivary gland.** Blood vessels are positive for ACE2 (original magnification x400).

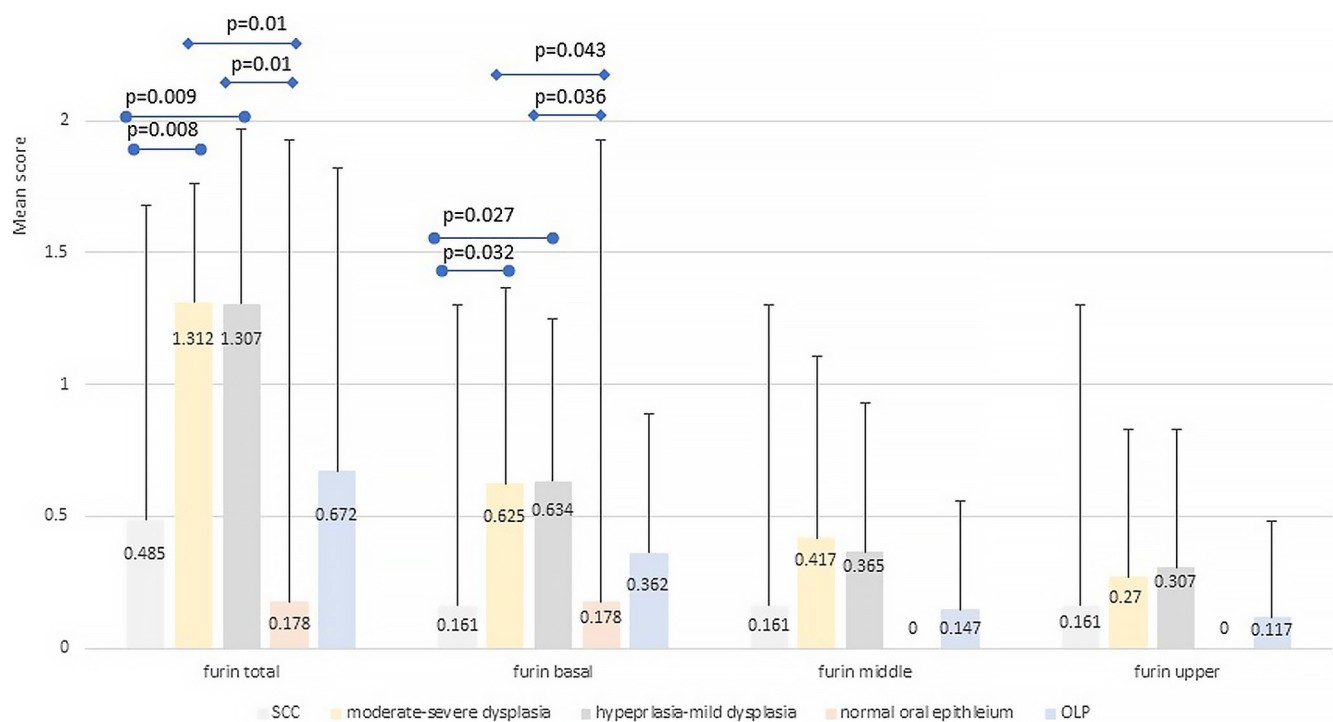

**Fig 4. Mean scores of furin expression.** Results show the various epithelial-related pathologies according to the mean total score and mean scores of each epithelial third–basal, middle, upper.

## Discussion

ACE2 and furin are factors involved in a plethora of signaling pathways associated with conditions ranging from homeostasis to inflammation to cancer. In particular, furin has been shown to be associated with development of OSCC. Recently, both factors were shown to play a

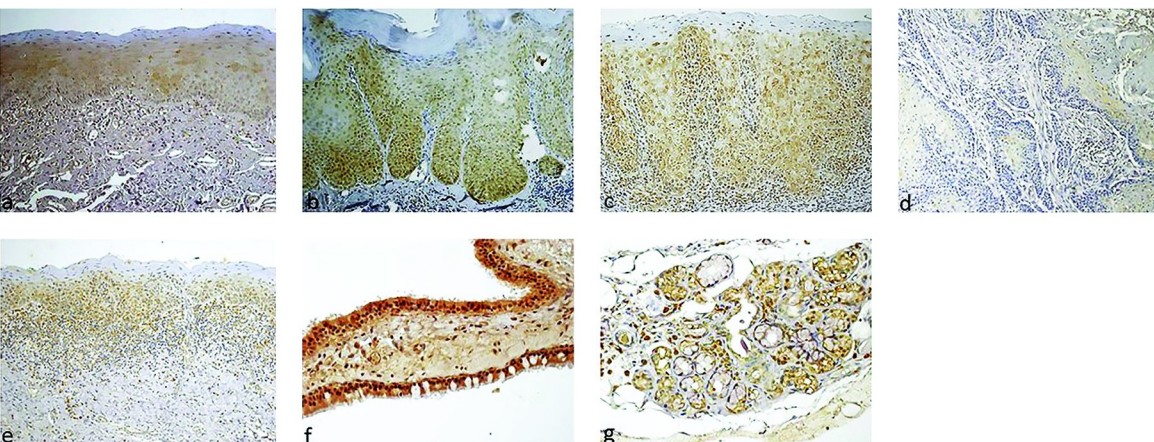

**Fig 5. Furin immunostaining in various oral tissues.** Normal control–total score 1 (basal 1—the most consistent stain, middle– 0, upper —0) (a), epithelial hyperplasia with mild dysplasia–total score 3 (score of 1 for each epithelial third) (b), epithelial hyperplasia with moderate-to-severe dysplasia–total score 4.5 (score of 1.5 for each epithelial third) (c), squamous cell carcinoma–total score 0 (d), oral lichen planus–total score 3 (basal, excluding the adjacent inflammatory reaction– 1.5, middle– 1.5, upper– 0) (e), upper respiratory tract mucosa–strong staining intensity of the respiratory epithelial cells (f), minor salivary glands—marked positive staining of the gland parenchyma (g) (a-e X200 original magnification, f-g original magnification x400).

major role in the attachment and infection of host cells by SARS-CoV-2. This role can only be fulfilled under the conditions that ACE2 and furin (and other co-factors) are concurrently expressed on the cell membrane of epithelial cells, and that these cells line the surface of a tract, through which the virus enters into the body. We aimed to investigate the expression of furin and ACE2 in a variety of oral epithelial lesions in order to find out whether they may potentially increase the risk of SARS-CoV-2 infection in patients diagnosed with these lesions. Results showed that the oral epithelium unequivocally lacked expression of ACE2. ACE2 was only expressed by the luminal aspect of upper respiratory tract epithelium and by some small blood vessels within the parenchyma of the submandibular salivary gland. Furin expression was cytoplasmic and not membranous and was present mainly in dysplastic lesions but not in OSCC. In terms of oral carcinogenesis, it may have a function in the development of oral pre-malignant lesions. In terms of SARS-CoV-2 infection, although its expression was positive, in the absence of ACE2, furin by itself cannot promote binding of the virus. Therefore, individuals diagnosed with pre-malignant lesions, OSCC or chronic inflammatory conditions like OLP, do not seem to be at a higher risk of oral infection by SARS-CoV-2 in comparison to those with healthy oral mucosae. On a more general note, our findings do not seem to support the wide concept that oral mucosae express the essential SARS-CoV-2 entry factors, and thus might not act as a port of entry to the virus. We will focus the discussion on topics related to these contradicting findings.

The gene that encodes ACE2 has been identified in 2000, and it is considered as the ACE homologue, with about 42% identical shared residues in their catalytic domain [24]. Originally, Northern blotting methodology was used to identify distribution of ACE2 in human tissues, which revealed expression in the heart, kidney and testis, consistent with a possible role in cardio-renal function. Shortly after, distribution of ACE2 was analyzed in 72 types of human tissues by quantitative RT-PCR, and compared to that of ACE [24]. That study revealed an unanticipated expression of ACE2 in the gastrointestinal tract, in addition to the ACE-positive cardio-renal-related tissues, with an overall ACE2 expression less ubiquitous than that of ACE. ACE2 transcripts in the oral and upper respiratory tract have not been investigated in that study, neither expression of ACE2 protein nor its sub-cellular localization. The next corner-stone study was by Hamming et al., [25] published after the 2003 SARS-CoV outbreak and identification of ACE2 as the receptor for the virus. That study investigated the immunohisto-chemical expression of ACE2 in tissues that were selected to represent organs and systems, where the SARS-CoV has been detected in human patients, including samples of oral mucosa, tissues of the upper respiratory tract, such as nasal mucosa, and the nasopharynx. These specific tissues were found to weakly express ACE2 in the basal layers but not in the superficial epithelial cells, facing external surfaces. Our immunohistochemical results demonstrating absence of expression of ACE2 in oral tissues are largely in line with the findings of Hamming et al., [25] and also with those of Sato et al., [5] in regard to tongue epithelium. In contrast, other immunohistochemical studies claimed for a positive expression of ACE2 protein that is suitable for the entrance of SARS-CoV-2 through the oral epithelium [12, 14, 26]. These discrepancies could be explained by the use of different antibodies with varying specificity and reproducibility [11]. This further emphasizes the importance of validating the immunoresponse of the used ACE2 antibodies in relation to positive golden standard tissues, as shown in the HPA (i.e., kidney tissue), and including in the study adequate control tissues (i.e., respiratory epithelium). Both measures were performed in the present study, which enabled us to conclude the lack of expression of ACE2 in the oral mucosae, however, these supporting measures were usually not provided in the other studies. These backup measures gain further relevance in view of the high homology between ACE and ACE2, and the more abundant expression of ACE, which should be taken in consideration when reporting positive ACE2

immunoreaction in tissues not expected to be positive, like the oral mucosae. One of the major advantages of immunohistochemistry is the identification of the type of cells that express the investigated protein and the sub-cellular localization, which have to be always considered in the context of its biological role. In regard to ACE2 and SARS-CoV-2, even though expression in basal/supra-basal cells or a nuclear location [12, 14, 26] could be a technically positive staining, it is neither relevant for virus infectivity nor represents a specific staining pattern [11], respectively. Furthermore, identification of ACE2 in the keratin layer [12, 15] does not appear to be biologically meaningful, as these are dead epithelial cells with non-selective membrane permeability [27], into which the virus can enter uncontrolled, but it definitely cannot replicate. Finally, the triad of factors required for SARS-CoV-2 entry, (ACE2 as the major receptor and co-factors furin and TMPRSS2) should co-localize simultaneously at the cell membrane of surface epithelial cells in order to permit effective infection of the host cells [11, 25, 28], but this was not the case in the literature [12, 14], neither in our study regarding the expression of ACE2 and furin.

We have performed RT-PCR for identification of ACE2 transcripts to support the immunohistochemical results, anticipating a good correlation between the two methods [11]. RNA level measurements and immunohistochemical results of our samples of oral mucosae and kidney samples were well correlated, with the former being negative by both methodologies and the latter positive, in accordance with the HPA datasets. The maxillary sinus/nasal mucosae were immunohistochemically positive but negative by RT-PCR, which is also in accordance with the HPA data. This inconsistency can be explained by the fact that the life span of mRNA molecules is short, measured in hours, in contrast to proteins, which have considerably longer life span [29]. The more advanced, precise, current PCR method–single cell sequencing [30], has also been used in the context of gene expression of ACE2 in the oral mucosa [10, 26, 31]. In some of these studies, oral epithelial cells were reported as positive for ACE2-RNA however, an in-depth analysis of the results has revealed low expression, which ranged between less than 0.52% (120 all cell types of a total of 22,969 cells, of which 1.19% epithelial cells) [31], 1% (unknown analyzed total cells) [10] and up to ~2% (~37 cells of a total of 1,843 cells) [26]. These clusters of positive epithelial cells were identified as basal and suprabasal types. Yet, in another study, it was clearly stated that ACE2 transcripts in head and neck mucosae, including the tongue, were sparse (~ 0%), with the exception of high expression in the ciliated and excretory epithelial cells [16]. Therefore, accurate interpretation of the reported findings might cast doubt on the oral mucosae being a likely portal of entry for SARS-CoV-2, as both ACE2-RNA transcripts and their functional protein products are not at all or barely expressed, which tends to be in accordance with our study results.

Concluding that oral mucosae express ACE2 and thus can be an important site for viral infection and transmission [10, 12, 25, 26], has to conform with biological principles and be supportive of the clinical findings. On a biological level, one can theoretically assume that ACE2-SARS-CoV-2 interaction could occur in the oral basal/suprabasal epithelial cells that would be infected and serve as the site for intracellular viral replication, however this would eventually lead to SARS-CoV-2-induced cell death [32]. This is expected to be reflected by morphological cellular changes, which are already apparent ~10 hours after infection [33, 34]. However, insofar as it has been illustrated in published studies, oral epithelial cells assumed to be infected, looked almost intact or displayed only vague cellular morphological changes [10, 35]. Furthermore, if one could assume that SARS-CoV-2 infected basal/suprabasal cells (where ACE2 was supposed to be expressed), would reach the surface of the oral epithelium within a short time (i.e., hours) and shed into the oral cavity/saliva and become a source of virus transmission, this also does not seem to stand within the timeframe that takes a basal epithelial cell to move through the epithelial width and shed, which is on average 5 days [36], by which time

infected epithelial cells would be expected to be dead. Collectively, it seems that the reported pattern of expression of SARS-CoV-2 entry factors in the oral mucosae, may not be biologically supported, and can therefore lead, if correctly interpreted, to be consistent with our results.

A wide range of clinical oral lesions was attributed to SARS-CoV-2 infection and it has been covered in a vast body of literature, which is beyond the aim of the present study [37, 38]. However, when data was compiled, selected and systematically analyzed it became obvious that lesions lacked pathognomonic features and were inconsistent so that the current published literature does not allow establishment of a direct link between the oral lesions and causation by SARS-CoV-2 infection, but seems that these were rather related to oral manifestations secondary to existing comorbidities, secondary infections, or the treatment given to combat COVID-19 disease [39–44]. Furthermore, the interesting finding of ACE2 isoforms that do not bind SARS-CoV-2 [45–48], means that identification of ACE2 in host cells does not automatically mean universal viral attachment and infection potential, and should be further analyzed on a personalized level for individual susceptibility for infection. Like the biological aspect, the clinical findings in COVID-19 patients do not seem to support the concept that the oral mucosae can serve as a primary site of entrance for SARS-CoV-2.

We were able to immunohistochemically identify expression of furin in most of the types of lesions that were included in the study, but the staining was usually weak, basal and cytoplasmic. Similar results were also reported by others [12, 26]. *Furin* gene expression by PCR was found to be higher in OSCC than in normal oral mucosa but not different from dysplasia [28]. In contrast, we found that furin protein was highest in dysplasia, with all other types of lesions exhibiting very low expression. These inconsistent findings necessitate more in-depth investigation on the role of furin in pre-malignant and malignant oral epithelial lesions. In the context of SARS-CoV-2, single cell sequencing performed on apparently normal oral mucosae 2 cm away from reactive or benign tumors, showed that among 13 cell subclusters identified, about 10% of all cells were positive for furin transcripts and approximately half of them (~90 cells) were epithelial cells [26]. Nevertheless, without the presence of membranous ACE2, furin by itself cannot make a substantial contribution to the entry of the virus into the oral epithelial cells.

In spite of lack of ACE2 expression in the oral mucosae reported in the present study, additional lines of research should be considered in an attempt to explain its identification in other studies. For example, ACE2 positive epithelial cells of upper respiratory tract (i.e., lining mucosae of the nasal cavity and maxillary sinus) can detach and reach the posterior third of the dorsal tongue by posterior nasal drip, or alternatively, infected cells can reach the oral cavity by spreading from the SARS-CoV-2-infected lower respiratory tract [25, 49]. Binding between the virus and these cellular fragments/desquamated cells may constitute only the first step of the interaction with the virus, and whether cleavage processes of the spike protein by furin and TMPRSS2 may follow, still needs to be proven. In addition, ACE2 has a soluble variant (sACE2), which is generated by cleavage of its ectodomain by a disintegrin and metalloprotease 17 (ADAM17) [5, 15, 50]. As sACE2 reaches the oral cavity/saliva it can serve as a binding site for the virus. The oral site/host cells, where the rest of the enzymatic cleavages may occur in order to promote virus entry and infection, should be further revealed.

Limitations of our study consist of the use of one single antibody each for the immunohistochemical identification of ACE2 and furin, although both calibration and quality control of their performance were done in a strict manner following golden standards. Use of the microarray technique could have enabled us inclusion of more tissue samples and of various concurrent different antibodies in a time-effective manner.

Collectively, our investigation of several types of oral epithelial lesions and normal mucosa revealed no expression of ACE2 protein nor its transcripts, and an irregular pattern of

expression of furin protein, mainly cytoplasmic and basally positioned. These seem to make the oral mucosae unfavorable primary targets for infection by SARS-CoV-2 as far as ACE2-furin-TMPRSS2 pathway entry is regarded, irrespective of the normal/inflammatory/dysplastic/neoplastic status of the lining epithelial cells. Further investigation is needed to bridge between the present results and those of previous studies, which reported on the key-role that oral mucosae play in the entry of SARS-CoV-2. The implications of this future research should be used for re-definition of SARS-CoV-2-related public health positions.

## Supporting information

**S1 File. Furin scores for each epithelial third (basal, middle, upper), total score per each case and mean (±SD) per each study group.**
(PDF)

## Author Contributions

**Conceptualization:** Osnat Grinstein-Koren, Marilena Vered, Ayelet Zlotogorski-Hurvitz.

**Investigation:** Osnat Grinstein-Koren, Michal Lusthaus, Hilla Tabibian-Keissar, Ron Ilatov, Marilena Vered, Ayelet Zlotogorski-Hurvitz.

**Project administration:** Michal Lusthaus, Marilena Vered.

**Resources:** Osnat Grinstein-Koren, Ilana Kaplan, Ayelet Zlotogorski-Hurvitz.

**Validation:** Amos Buchner, Ayelet Zlotogorski-Hurvitz.

**Visualization:** Ayelet Zlotogorski-Hurvitz.

**Writing – original draft:** Osnat Grinstein-Koren, Ilana Kaplan, Marilena Vered, Ayelet Zlotogorski-Hurvitz.

**Writing – review & editing:** Ilana Kaplan, Amos Buchner.

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
