## [Decision Letter · Decision Letter 0]

20 Nov 2023

PONE-D-23-28297Pathological changes in oral epithelium and the expression of SARS-CoV-2 entry receptors, ACE2 and furinPLOS ONE

Dear Dr. Vered,

Thank you for submitting your manuscript to PLOS ONE. After careful consideration, we feel that it has merit but does not fully meet PLOS ONE’s publication criteria as it currently stands. Therefore, we invite you to submit a revised version of the manuscript that addresses the points raised during the review process.

We look forward to receiving your revised manuscript.

Kind regards,

Cheorl-Ho Kim, Ph.D.

Academic Editor

PLOS ONE

Journal Requirements:

Additional Editor Comments:

Dear Dr Vered

Thank you for your kind submission od your study to PLOS One.

I have completed the scientific evaluation of your study.

I feel that your study is interesting in our readers, as you have emphasized. However, the general aspects of the experimental approach are not satified with our experts. Please note that your results are contrasted to previously reported findings, as you have also mentioned. Other criticisms are still in editorial concerns. However, I would like to receive your revision with interests.

For revision, the criticisms have been attached below.

Among them, the following critisims are carefully viewed:

Issues of Protein and RNA degradation in FFPE samples and low sensitivity to ACE2 expression detection. All samples were negative with the exception of those from the upper respiratory mucosa.

You have formulated the conclusion of Oral mucosa, normal or with epithelial pathologies lacks ACE2 expression.

In Nat Med 2021 (Ref 3), ACE2 expression was detected in nine oral epithelial clusters. However, you describe that ACE2-positive oral epithelial cells were rare. Please remind that Multiple oral epithelial cell subtypes are susceptible to SARS-CoV-2 infection in previous reports.

Additionally, two reviws are also helpful for your better revision.

Your revision would be re-reviewed.

Thank you

Sincerely

Cheorl-Ho Kim PhD Professor

SKKU Biological Science Dept

Korea

Editor of Plos One

Reviewers' comments:

Reviewer's Responses to Questions

**Comments to the Author**

1. Is the manuscript technically sound, and do the data support the conclusions?

Reviewer #1: No

Reviewer #2: Yes

Reviewer #3: Yes

2. Has the statistical analysis been performed appropriately and rigorously? 

Reviewer #1: N/A

Reviewer #2: I Don't Know

Reviewer #3: N/A

3. Have the authors made all data underlying the findings in their manuscript fully available?

Reviewer #1: No

Reviewer #2: Yes

Reviewer #3: Yes

4. Is the manuscript presented in an intelligible fashion and written in standard English?

Reviewer #1: Yes

Reviewer #2: Yes

Reviewer #3: Yes

5. Review Comments to the Author

Reviewer #1: The aim of the present study was to analyze the expression of ACE2 and furin in a total of 169 oral

pre-malignant lesions with different degrees of dysplasia and oral cancer, as well as oral inflammatory lesions represented by oral lichen planus (OLP), in order to investigate whether these classes of patients are potentially at a higher risk of infection by SARS-CoV-2. (Lines 68-70). Gene expression of ACE2 was measured by real-time RT- PCR on the total RNA extracted from formalin-fixed, paraffin-embedded (FFPE) tissue samples. Moreover, immunohistochemical staining of ACE2 and furin was performed.

The paper is well-written and methodologies are sound; however, I would suspect protein and RNA degradation in FFPE samples and/or a low sensitivity to ACE2 expression detection, because all samples were negative, with the exception of those from the upper respiratory mucosa. On the base of these negative results, the authors formulate the conclusion “Oral mucosa, normal or with epithelial pathologies lacks ACE2 expression” and “

• These results contrast with several published papers. As an example, PMID: 33767405 (Nat Med 2021 that is ref 3 in this paper) reported that “ ACE2 expression was detected in nine oral epithelial clusters, including Basal 1–3, basal cycling, SG ducts, SG serous and SG mucous acini clusters.” The authors state in Discussion (lines 251-252) “In these studies (ref 3, 20 and 21) ACE2-positive oral epithelial cells were rare, comprised only ~1% of all identified types of cells (ranging from tens to a few hundreds of positive cells) and limited mainly to the basal/suprabasal localization” Even so, the paper cited in ref 3 (and many others) concluded that “These findings suggest that multiple oral epithelial cell subtypes are susceptible to (SARS-CoV-2) infection.”

Samples from biopsies of amalgam tattoo were considered “normal controls”. I’m not an expert in pathology, but I retain that amalgam tattoo cannot be considered a good negative control for the oral epithelium as they are lesions. The lack of non-pathologic oral tissues should be considered among the study limitations and a possible reason for the negative results for ACE2 expression different from other studies. Accordingly, the abstract Conclusions: Oral mucosa, normal or with epithelial pathologies lacks ACE2 expression, while furin is weak and basal only, suggesting alternative entry routes for SARS-CoV-2 should be investigated.” are not substantiated because a normal oral mucosa was not tested.

Other points to be addressed:

Introduction: line 62 “In view of furin being part of the triad of SARS-CoV-2 entry receptors, it may be... “

This statement should be amended as furin is not a SARS-CoV-2 entry receptor, but allows protease cleavage during one of the options for entry into host cells.

Table 1: RT-PCR study outcomes for gene expression of ACE2. The data in this table only reproduce the text. It would be more informative to provide the results of the 2-ΔΔCt measurement.

Reviewer #2: Reviewer Report:

Manuscript Title: Pathological changes in oral epithelium and the expression of SARS-CoV-2 entry

receptors, ACE2 and furin

Manuscript ID: PONE-D-23-28297

Introduction: The introduction of your manuscript provides a comprehensive background on the role of the oral cavity in COVID-19 pathogenesis, particularly focusing on the roles of furin and ACE2. However, there are several areas where improvements could significantly enhance the clarity and impact of your introduction:

1. Explicit Hypothesis: The introduction would benefit from stating a clear hypothesis that links furin's role in oral cancer and ACE2 in inflammatory conditions to an increased risk of SARS-CoV-2 infection. This would provide a more direct connection between the background information and the study's aims.

2. Streamlining Content: The introduction covers a broad range of topics. Streamlining these points to more directly lead into the study's focus would enhance clarity and relevance.

3. Connection Between Topics: The introduction discusses furin's role in both SARS-CoV-2 entry and oral cancer but does not explicitly make the connection between these two roles. Clarifying how furin's function in cancer might influence SARS-CoV-2 susceptibility would strengthen the rationale for the study.

4. Background Context: A brief contextualization of the current understanding of COVID-19's interaction with the oral cavity at the beginning of the introduction would set a more comprehensive stage for the specific focus on furin and ACE2.

5. Clarification of Terms: Some terms and concepts, such as the role of TMPRSS2 and the mechanism of viral entry into cells, are mentioned but not fully explained. A brief elaboration on these points could make the introduction more accessible to readers not familiar with the subject.

6. Reference to Current Literature: Include more current research or reviews that support the evolving understanding of the oral cavity's role in COVID-19 transmission and pathogenesis. For example, Khirfan et al. (2022) and J Al-Awaida et al. (2021) provide insights into the molecular interactions and epidemiological correlates of SARS-CoV-2.

7. Balancing Detail and Brevity: The introduction is detailed in areas like the role of furin in cancer. Balancing these details with a more concise presentation could improve readability and focus.

8. Link to Study Aim: More explicitly link the background information to the rationale for analyzing ACE2 and furin in oral lesions to make the introduction more cohesive.

9. Please add following references for your introduction

Khirfan F, Jarrar Y, Al-Qirim T, Goh KW, Jarrar Q, Ardianto C, Awad M, Al-Ameer HJ, Al-Awaida W, Moshawih S, Ming LC. Analgesics induce alterations in the expression of SARS-CoV-2 entry and arachidonic-acid-metabolizing genes in the mouse lungs. Pharmaceuticals. 2022 Jun 1;15(6):696.

J Al-Awaida W, Jawabrah Al Hourani B, Swedan S, Nimer R, Alzoughool F, J Al-Ameer H, E Al Tamam S, Alashqar R, Gushchina Y, Samy Abousenna M, Ayyash AM. Correlates of SARS-CoV-2 Variants on Deaths, Case Incidence and Case Fatality Ratio among the Continents for the Period of 1 December 2020 to 15 March 2021. Genes. 2021 Jul;12(7):1061.

Materials and Methods: Your materials and methods section is detailed and follows STROBE reporting standards. However, certain improvements could enhance its clarity and reproducibility:

1. Sample Selection Criteria: Provide more specific information about the "key-words of diagnostic entities" used for selecting cases.

2. Control Group Selection: Expand on why upper respiratory mucosa tissues are appropriate controls for the study.

3. Staining and Scoring System Clarity: Simplify the explanation of the histomorphometric analysis and scoring system or provide a visual aid.

4. Details on RT-PCR Procedure: Clarify certain steps, such as the choice of the number of tissue sections and the rationale behind the minor modifications to the RNA extraction protocol.

5. Statistical Analysis Specifics: Provide more details on the assumptions checked before applying One-way ANOVA with Bonferroni correction, and justify the choice of this specific statistical test.

6. Ethical Approval Clarity: Include a brief statement on how participant confidentiality and data privacy were maintained.

7. Reference to Standards and Protocols: Mention any other relevant guidelines or protocols followed during the study.

8. Handling of Potential Biases: Discuss how potential biases were addressed or minimized.

9. Explanation of Figure 1: Include a brief description of what Figure 1 illustrates.

10. Detailing RNA Concentration Measurements: Include details about the range of acceptable RNA concentrations and how variations were handled.

Discussion: The discussion section of your manuscript presents an insightful analysis of your findings. However, there are areas where improvements could further enhance the depth and impact of your discussion:

1. Interpretation of Negative Results: Further explore the implications of the negative findings regarding ACE2 expression in oral epithelial cells, particularly in terms of their significance in the broader context of COVID-19 research and oral health.

2. Comparison with Other Studies: Provide a more detailed analysis of how your results align or diverge from existing literature, and discuss the possible reasons for any discrepancies.

3. Speculation and Hypothesis Generation: Clearly frame speculative insights as hypotheses for future research rather than conclusions.

4. Clarification of Methodological Limitations: Discuss the potential impact of limitations, such as the variability in antibodies used across studies, on the study's findings.

5. Broader Contextualization: Situate the study within the larger body of COVID-19 research, particularly concerning the transmission dynamics of the virus and the role of the oral cavity.

6. Discussion of Clinical Implications: Expand on how these findings might influence clinical practice or public health measures.

7. Future Research Directions: Explicitly outline potential directions for future research, especially in terms of exploring the role of furin and ACE2 in other aspects of COVID-19 pathogenesis and transmission.

8. Statistical Interpretation: Discuss how the statistical findings support (or don’t support) the study’s conclusions.

9. Addressing Contradictory Evidence: Provide a more comprehensive discussion of why contradictions with existing literature might exist and what they imply for the field.

10. Integration of Molecular and Clinical Perspectives: Integrate the molecular findings with clinical observations and implications for a more holistic understanding of the topic.

Reviewer #3: The reviewer appreciates the submission and has a few suggestions to improve the manuscript.

1) The lack of ACE2 receptors in squamous epithelium and the inconsistent furin expression has already been published, but the role of macrophages remains unclear and needs to be discussed. The detection of macrophages is somehow laborious in the given biopsies, but it should be added.

2) It would be of great interest to stain mucosal samples from patients with SARS Cov 2 Infection, maybe ssome can be obtained from autoptic material.

3) The figures with the immunohistochemical stainings need to be improved.

4) Literature is missing: only two examples.....

Determinants of SARS-CoV-2 entry and replication in airway mucosal tissue and susceptibility in smokers.

Nakayama T et al. Cell Rep Med. 2021 Oct 19;2(10):100421. doi: 10.1016/j.xcrm.2021.100421. Epub 2021 Sep 28.

PMID: 34604819

Comprehensive analysis of SARS-CoV-2 receptor proteins in human respiratory tissues identifies alveolar macrophages as potential virus entry site.

Bräutigam K, Reinhard S, Wartenberg M, Forster S, Greif K, Granai M, Bösmüller H, Klingel K, Schürch CM.

Histopathology. 2023 May;82(6):846-859. doi: 10.1111/his.14871. Epub 2023 Feb 21.

PMID: 36700825

6. PLOS authors have the option to publish the peer review history of their article (what does this mean?). If published, this will include your full peer review and any attached files.

Reviewer #1: No

Reviewer #2: No

Reviewer #3: No

---

## [Author Response · Author response to Decision Letter 0]

14 Dec 2023

The below responses to Reviewers' comments have also been uploaded as a separated file, as required. As it is a long document including text, figure and table, it might be clearer to read the uploaded document.

PONE-D-23-28297

Pathological changes in oral epithelium and the expression of SARS-CoV-2 entry receptors, ACE2 and furin

PLOS ONE

Dear Dr. Vered,

Thank you for submitting your manuscript to PLOS ONE. After careful consideration, we feel that it has merit but does not fully meet PLOS ONE’s publication criteria as it currently stands. Therefore, we invite you to submit a revised version of the manuscript that addresses the points raised during the review process.

If applicable, we recommend that you deposit your laboratory protocols in protocols.io to enhance the reproducibility of your results. Protocols.io assigns your protocol its own identifier (DOI) so that it can be cited independently in the future. For instructions see: https://journals.plos.org/plosone/s/submission-guidelines#loc-laboratory-protocols. Additionally, PLOS ONE offers an option for publishing peer-reviewed Lab Protocol articles, which describe protocols hosted on protocols.io. Read more information on sharing 

protocols at https://plos.org/protocols?utm_medium=editorial-email&utm_source=authorletters&utm_campaign=protocols.

We look forward to receiving your revised manuscript.

Kind regards,

Cheorl-Ho Kim, Ph.D.

Academic Editor

PLOS ONE

Journal Requirements:

Response: Uploaded files were named as required

Response: Required details were added in Material and Methods, p. 5, lines 187-191.

Response: The information on financial support is "The study was supported by the Herb and Ed Stein Chair in Oral Pathology, Tel Aviv University". This has no grant number nor could it be identified among those foundations registered in the submission site.

Response: The study does not contain blot/gel results.

Response: ORCID iD was updated.

 Response: Caption was added, p. 25, lines 1068-1070; reference to this file in the text was done on p. 9, line 337.

Additional Editor Comments:

Dear Dr Vered

Thank you for your kind submission od your study to PLOS One.

I have completed the scientific evaluation of your study.

I feel that your study is interesting in our readers, as you have emphasized. However, the general aspects of the experimental approach are not satified with our experts. Please note that your results are contrasted to previously reported findings, as you have also mentioned. Other criticisms are still in editorial concerns. However, I would like to receive your revision with interests.

For revision, the criticisms have been attached below.

Among them, the following critisims are carefully viewed:

Issues of Protein and RNA degradation in FFPE samples and low sensitivity to ACE2 expression detection. All samples were negative with the exception of those from the upper respiratory mucosa.

You have formulated the conclusion of Oral mucosa, normal or with epithelial pathologies lacks ACE2 expression.

In Nat Med 2021 (Ref 3), ACE2 expression was detected in nine oral epithelial clusters. However, you describe that ACE2-positive oral epithelial cells were rare. Please remind that Multiple oral epithelial cell subtypes are susceptible to SARS-CoV-2 infection in previous reports.

Response: We thank the Editor for this remark. We have thoroughly analyzed the results of this well-written and comprehensive study by Huang et al. (original reference #3, now #10). In particular, we looked at Fig. 2d (please see below), which illustrates the single cell RNA-seq in 9 types of epithelial cells that were investigated for the expression of different receptors/factors that can be involved in the attachment of and infection by SARS-CoV-2. These 9 types of epithelial cells comprised of 3 basal, and one type each of cycling, suprabasal, ductal, serous, mucous and myoepithelial cells, where the first five types originate from the oral mucosa while the last four types clearly originate from salivary glands. It should be noted that this analysis did not include any cell type higher than suprabasal. None of the studied oral mucosa cells (basal, cycling and suprabasal) can be directly accessed by the virus, which 

can only bind to receptors at the epithelial surface. The frequency of expression of ACE2 in the oral mucosa cells was found only in the 3 basal and the cycling cell types, and it was as low as ~1% of the cells; the suprabasal cells were totally negative for the expression of ACE2. Fig. 2f (please see below) shows the frequency of expression of ACE2 after normalization and even then, the frequency of expression of ACE2 in the basal and cycling cell types from the oral mucosa remained minute. It is only when comparing this extremely low expression of ACE2 in the oral mucosa cells to the high expression (both frequency and intensity) found in the respiratory nasal epithelium and intestinal epithelium, that we really can appreciate what a genuine high expression of ACE2 is. So, one can conclude from these findings, that even the most sensible technique for the detection of expression of ACE2 analyzed per cell types of oral mucosa, did not reveal a genuine proven and significant expression of ACE2, although the authors considered this as a "positive" expression. 

Attentive reading of an additional study that also used single cell RNA-seq (original reference #20, now #31), analyzed 4 oral tissues and acquired and identified 7 types of cells, including epithelial, fibroblasts, T cells, macrophages, mast cells, B cells and endothelial cells – a total of 22,969 cells. The authors stated that "we confirmed the ACE2 was expressed in oral tissues (0.52% ACE2-positive cells)", which means that among all the 7 types of cells (!), roughly 120 cells were positive for the expression of ACE2. The authors then specifically address to the epithelial cells: "ACE2-positive cells could be found in oral tissues including epithelial cells (1.19% ACE2-positive cells)", which is not entirely clear whether the 1.19% is from the total 120 ACE2-positive oral tissue cells (which means ~one positive cell), or from the total number of examined cells (i.e., 22,969), in which case it means 273 ACE2-positive epithelial cells [this does not make much sense as the authors previously stated that 0.52% (120 cells) of ALL ORAL TISSUES were ACE2 positive]. Figure 2d and 2e, which are supposed to illustrate the ACE2 results, shows no real violin plots and figure 2e, which is supposed to illustrate UMAP distribution of ACE2 expression is in gray and actually has no scale for expression intensity. Taken together these results and their way of illustration might cast doubt on the authors' claim for "High expression of ACE2 receptor of 2019-nCoV on the epithelial cells of oral mucosa".

Yet, another study by Zhong et al., (original reference #21, now #26) analyzed a total of 1,843 cells isolated from the oral mucosa of five patients using single cell RNA-sequence. They stated that "Then, we calculated the percentage of ACE2-expressing cells in the dataset (Figure 2E). We find that, in oral tissue, the percentage of ACE2-positive cells was 2.2%. Among them, 92% were epithelial cells.". Calculating 2.2% out of 1,843 total analyzed cells, means that 40 cells from the oral tissues were ACE2 positive, and 92% of these means ~37 cells being ACE2 positive epithelial cells!!!

We may conclude that a very attentive reading of the results can uncover a more accurate information on the data, which is not always in accordance with the more general conclusions and titles of those studies. We chose not to show these detailed calculations in the text of the manuscript, in respect of the authors, but just made a general note on what the reported % could mean in absolute numbers of cells – p. 14, lines 553-563.

Additionally, two reviws are also helpful for your better revision.

Your revision would be re-reviewed.

Thank you

Sincerely

Cheorl-Ho Kim PhD Professor

SKKU Biological Science Dept

Korea

Editor of Plos One

Reviewers' comments:

Reviewer's Responses to Questions

Comments to the Author

1. Is the manuscript technically sound, and do the data support the conclusions?

Reviewer #1: No

Reviewer #2: Yes

Reviewer #3: Yes

2. Has the statistical analysis been performed appropriately and rigorously? 

Reviewer #1: N/A

Reviewer #2: I Don't Know

Reviewer #3: N/A

3. Have the authors made all data underlying the findings in their manuscript fully available?

Reviewer #1: No

Reviewer #2: Yes

Reviewer #3: Yes

4. Is the manuscript presented in an intelligible fashion and written in standard English?

Reviewer #1: Yes

Reviewer #2: Yes

Reviewer #3: Yes

5. Review Comments to the Author

Reviewer #1: The aim of the present study was to analyze the expression of ACE2 and furin in a total of 169 oral pre-malignant lesions with different degrees of dysplasia and oral cancer, as well as oral inflammatory lesions represented by oral lichen planus (OLP), in order to investigate whether these classes of patients are potentially at a higher risk of infection by SARS-CoV-2. (Lines 68-70). Gene expression of ACE2 was measured by real-time RT- PCR on the total RNA extracted from formalin-fixed, paraffin-embedded (FFPE) tissue samples. Moreover, immunohistochemical staining of ACE2 and furin was performed.

The paper is well-written and methodologies are sound; however, I would suspect protein and RNA degradation in FFPE samples and/or a low sensitivity to ACE2 expression detection, because all samples were negative, with the exception of those from the upper respiratory mucosa. On the base of these negative results, the authors formulate the conclusion “Oral mucosa, normal or with epithelial pathologies lacks ACE2 expression” and “

• These results contrast with several published papers. As an example, PMID: 33767405 (Nat Med 2021 that is ref 3 in this paper) reported that “ ACE2 expression was detected in nine oral epithelial clusters, including Basal 1–3, basal cycling, SG ducts, SG serous and SG mucous acini clusters.” The authors state in Discussion (lines 251-252) “In these studies (ref 3, 20 and 21) ACE2-positive oral epithelial cells were rare, comprised only ~1% of all identified types of cells (ranging from tens to a few hundreds of positive cells) and limited mainly to the basal/suprabasal localization” Even so, the paper cited in ref 3 (and many others) concluded that “These findings suggest that multiple oral epithelial cell subtypes are susceptible to (SARS-CoV-2) infection.”

Response: We thank the reviewer for raising these important points. We also confronted these challenging points while planning the design of the study as well as along the "Discussion" and attempted to provide sound support for our findings, as follows:

1. Our results on lack of expression of ACE2 in the oral epithelium and positive expression in the respiratory epithelium are in accordance with those of the golden standard in the Human Protein Atlas (Fig. 1) 

2. All tissue samples have undergone the same routine fixation, embedding and processing procedures in our laboratory. We assume that if there were problems with the tissues, such as protein and/or RNA degradation with resulting lack of detection of the ACE2 protein or its mRNA, it should have been valid also for tissues which we used to calibrate the ACE2 antibody (samples of kidney tissue), control tissues of upper respiratory tract, and intrinsic control of small blood vessels within salivary glands, which were occasionally present in the some of the oral biopsies – all these internal and external controls, were found to be ACE2 positive. The integrity of the RNA was proven by the positive signal of the household gene GAPDH also in those samples where expression of ACE2 transcript was negative (please see below, p. 14 - our response to point #10) 

3. In regard to the findings reported in published studies (original references #3, 20; now #10 and 31, respectively), we have already responded above to a similar comment of the Editor, please see pp. 4-5. Regarding reference #20 (now reference #31): authors reported only a 2.2% expression of ACE2 in all types of oral cells by using single cell RNA-seq. This percentage seems to be markedly low when taking into consideration that the analysis was run on a substantial amount of 12 oral mucosa samples. Upon looking at the immunohistochemical photomicrographs, it is clear that the expressed ACE2 protein is located in the basal and some of the suprabasal epithelial cells, mostly in their cytoplasm, whereas, it is important to emphasize again that in order to serve as a port of entry for the SARS-CoV-2 virus its expression must be on the most superficial epithelial cells and it must involve the outer cell membranes, not the cytoplasm 

Samples from biopsies of amalgam tattoo were considered “normal controls”. I’m not an expert in pathology, but I retain that amalgam tattoo cannot be considered a good negative control for the oral epithelium as they are lesions. The lack of non-pathologic oral tissues should be considered among the study limitations and a possible reason for the negative results for ACE2 expression different from other studies. Accordingly, the abstract Conclusions: Oral mucosa, normal or with epithelial pathologies lacks ACE2 expression, while furin is weak and basal only, suggesting alternative entry routes for SARS-CoV-2 should be investigated.” are not substantiated because a normal oral mucosa was not tested.

Response: Amalgam tattoo is a subepithelial pigmentation caused by the iatrogenic implantation of tiny amalgam particles, which usually remain inert and cause no inflammatory reaction, therefore has no impact on the integrity of the oral lining epithelium. In general, there is no definite "normal" oral mucosae devoid of any change or modification due to the fact that these are being permanently exposed to internal and external irritant factors. In comparison, we could question the use of oral mucosa samples in other studies, in particular, those that included gingival tissues, which are always involved by inflammation. This often introduce modifications in the overlying gingival epithelium and one should wonder if this might not affect the expression/over-expression/no-expression of ACE2 in the gingival tissues (e.g., original references #5, 20, 21, now #12, 31, 26, respectively). Under these circumstances, even if not "perfect", amalgam tattoo seems to be as close to normal oral mucosae, as possible. 

Other points to be addressed:

Introduction: line 62 “In view of furin being part of the triad of SARS-CoV-2 entry receptors, it may be... “

This statement should be amended as furin is not a SARS-CoV-2 entry receptor, but allows protease cleavage during one of the options for entry into host cells.

Response: Corrections were done throughout the text, as suggested by the Reviewer (p. 3, line 119; p. 4 line 128; p. 11 line 379; p. 13, line 514).

Table 1: RT-PCR study outcomes for gene expression of ACE2. The data in this table only reproduce the text. It would be more informative to provide the results of the 2-ΔΔCt measurement.

Response: Since the various analyzed study samples had mean Ct of a value of 0 regarding expression of the ACE2 gene except for the kidney, the 2^-ΔΔCt had no biological relevance, therefore under these circumstances, we decided only to mention this in the text itself and to delete Table 1 (p. 10, lines 365-6). 

Reviewer #2: Reviewer Report:

Manuscript Title: Pathological changes in oral epithelium and the expression of SARS-CoV-2 entry

receptors, ACE2 and furin

Manuscript ID: PONE-D-23-28297

Introduction: The introduction of your manuscript provides a comprehensive background on the role of the oral cavity in COVID-19 pathogenesis, particularly focusing on the roles of furin and ACE2. However, there are several areas where improvements could significantly enhance the clarity and impact of your introduction:

Response to points #1-5, 7, 8 – We thank the reviewer for the constructive comments. As a result, Introduction section has been almost entirely re-written, including substantial addition of relevant literature (pp. 3-5, lines 102-182).

1. Explicit Hypothesis: The introduction would benefit from stating a clear hypothesis that links furin's role in oral cancer and ACE2 in inflammatory conditions to an increased risk of SARS-CoV-2 infection. This would provide a more direct connection between the background information and the study's aims.

2. Streamlining Content: The introduction covers a broad range of topics. Streamlining these points to more directly lead into the study's focus would enhance clarity and relevance.

3. Connection Between Topics: The introduction discusses furin's role in both SARS-CoV-2 entry and oral cancer but does not explicitly make the connection between these two roles. Clarifying how furin's function in cancer might influence SARS-CoV-2 susceptibility would strengthen the rationale for the study.

4 Background Context: A brief contextualization of the current understanding of COVID-19's interaction with the oral cavity at the beginning of the introduction would set a more comprehensive stage for the specific focus on furin and ACE2.

5 Clarification of Terms: Some terms and concepts, such as the role of TMPRSS2 and the

mechanism of viral entry into cells, are mentioned but not fully explained. A brief elaboration on these points could make the introduction more accessible to readers not familiar with the subject.

6. Reference to Current Literature: Include more current research or reviews that support the evolving understanding of the oral cavity's role in COVID-19 transmission and pathogenesis. For example, Khirfan et al. (2022) and J Al-Awaida et al. (2021) provide insights into the molecular interactions and epidemiological correlates of SARS-CoV-2.

Response: There is a considerable wealth of literature on COVID-19 transmission and pathogenesis, in general, and regarding the role of the oral cavity, in particular. In order to keep the Introduction focused, we have chosen to cite those that investigated expression of entry factors of the oral mucosa in healthy subjects mainly using immunohistochemistry. The suggested study by Khirfan et al., is very interesting, however although it investigated entry factors for SARS-CoV-2, it focused on lung tissues in a mouse model and involved treatment modalities, therefore we felt that it was not entirely relevant to the present study. The other suggested study by Al-Awaida et al., was introduced as reference #1. Other references were added according to additional changes performed in the Introduction (references #2-7). 

7. Balancing Detail and Brevity: The introduction is detailed in areas like the role of furin in cancer. Balancing these details with a more concise presentation could improve readability and focus.

8. Link to Study Aim: More explicitly link the background information to the rationale for analyzing ACE2 and furin in oral lesions to make the introduction more cohesive.

9. Please add following references for your introduction

Khirfan F, Jarrar Y, Al-Qirim T, Goh KW, Jarrar Q, Ardianto C, Awad M, Al-Ameer HJ, Al-Awaida W, Moshawih S, Ming LC. Analgesics induce alterations in the expression of SARS-CoV-2 entry and arachidonic-acid-metabolizing genes in the mouse lungs. Pharmaceuticals. 2022 Jun 1;15(6):696.

J Al-Awaida W, Jawabrah Al Hourani B, Swedan S, Nimer R, Alzoughool F, J Al-Ameer H, E Al Tamam S, Alashqar R, Gushchina Y, Samy Abousenna M, Ayyash AM. Correlates of SARS-CoV-2 Variants on Deaths, Case Incidence and Case Fatality Ratio among the Continents for the Period of 1 December 2020 to 15 March 2021. Genes. 2021 Jul;12(7):1061.

Response: We addressed this comment in our response to point #6.

Materials and Methods: Your materials and methods section is detailed and follows STROBE reporting standards. However, certain improvements could enhance its clarity and reproducibility:

1. Sample Selection Criteria: Provide more specific information about the "key-words of diagnostic entities" used for selecting cases.

Response: Diagnostic entities were detailed on pp. 5-6, lines 201-216.

2. Control Group Selection: Expand on why upper respiratory mucosa tissues are appropriate controls for the study. 

Response: Explanation was added (p. 6, lines 216-217).

3. Staining and Scoring System Clarity: Simplify the explanation of the histomorphometric analysis and scoring system or provide a visual aid.

Response: The histomorphometric analysis was further detailed in a simplified manner (pp. 6-7, lines 235-254). The legend of Fig 5 now also provides the scores attributed to the illustrated samples (p. 10, lines 352-360)

4. Details on RT-PCR Procedure: Clarify certain steps, such as the choice of the number of tissue sections and the rationale behind the minor modifications to the RNA extraction protocol.

Response: As the immunohistochemical stain for ACE2 in all oral tissues was negative, we submitted to the PCR study representative cases from the study groups + control tissues, selecting especially those cases which contained a satisfactory amount of tissue within the paraffin blocks (p. 7, lines 256-257). 

The main minor change in RNA extraction protocol was elongating the incubation time with proteinase K to 2-4 h, the rationale being to increase tissue digestion and RNA recovery (p. 7, lines 262-264).

5. Statistical Analysis Specifics: Provide more details on the assumptions checked before applying One-way ANOVA with Bonferroni correction, and justify the choice of this specific statistical test.

Response: Addition of details was performed (p. 8, lines 299-302).

6. Ethical Approval Clarity: Include a brief statement on how participant confidentiality and data privacy were maintained.

Response: Explanation was added (p. 5, lines 187-199).

7. Reference to Standards and Protocols: Mention any other relevant guidelines or protocols followed during the study.

Response: Immunohistochemical stains for ACE2 and furin were performed as recommended by the manufacturers of these commercial antibodies – catalog numbers of the used antibodies were provided (p. 6, lines 221-224). 

PCR – RNA is routinely extracted from formalin fixed paraffin embedded tissues in our molecular pathology lab for diagnostic tests according to the protocol used for identification of KIAA-BRAF fusion for diagnosis of pilocytic astrocytoma, as described by Tian et.al., reference #22, and research projects by the Qiagen columns (RNeasy, cat.#, available at www.qiagen.com/us/resources/download.aspx?id=20564fa0-0a23-4e35-971d-b403f69aa39a&lang=en) (p. 7, lines 255-259).

8. Handling of Potential Biases: Discuss how potential biases were addressed or minimized.

Response: Point was addressed (p. 7, lines 253-254). 

9. Explanation of Figure 1: Include a brief description of what Figure 1 illustrates.

Response: Description of Fig 1 was added in the caption (p. 6, lines 229-234).

10. Detailing RNA Concentration Measurements: Include details about the range of acceptable RNA concentrations and how variations were handled.

Response: RNA concentration is measured by spectrophotometer (NanoDropTM 1000). 1000ng of RNA (in a volume of 13.2ul) is usually used for reverse transcription reaction in order to amplify cDNA by the High-Capacity cDNA Reverse Transcription Kit of Applied Biosystems (cat. no. 4368814) according to manufacturer protocol. The minimal accepted amount of RNA for cDNA is around 250ng. In the current research, 1000 ng RNA was used for cDNA amplification in all samples. This was mentioned, pp. 7-8, lines 267-291).

In regard to variations in RNA concentrations, we were able to establish that variations were minimal as the mean Ct of GAPDH for all tested samples were in in the same range, as can be seen below:

 GAPDH - mean Ct

Hyperplasia/mild dysplasia 31.29033

Hyperplasia/mild dysplasia 32.51602

OLP 35.40825

SCC 29.49092

Mod-severe dysplasia 33.46436

SCC 34.01926

Amalgam tattoo 35.84667

Nasal mucosa 31.09767

Kidney 29.05574

Maxillary sinus 36.59096

Discussion: The discussion section of your manuscript presents an insightful analysis of your findings. However, there are areas where improvements could further enhance the depth and impact of your discussion:

1. Interpretation of Negative Results: Further explore the implications of the negative findings regarding ACE2 expression in oral epithelial cells, particularly in terms of their significance in the broader context of COVID-19 research and oral health.

2. Comparison with Other Studies: Provide a more detailed analysis of how your results align or diverge from existing literature, and discuss the possible reasons for any discrepancies.

3. Speculation and Hypothesis Generation: Clearly frame speculative insights as hypotheses for future research rather than conclusions.

4. Clarification of Methodological Limitations

: Discuss the potential impact of limitations, such as the variability in antibodies used across studies, on the study's findings.

5. Broader Contextualization: Situate the study within the larger body of COVID-19 research, particularly concerning the transmission dynamics of the virus and the role of the oral cavity.

6. Discussion of Clinical Implications: Expand on how these findings might influence clinical practice or public health measures.

7. Future Research Directions: Explicitly outline potential directions for future research, especially in terms of exploring the role of furin and ACE2 in other aspects of COVID-19 pathogenesis and transmission.

8. Statistical Interpretation: Discuss how the statistical findings support (or don’t support) the study’s conclusions.

9. Addressing Contradictory Evidence: Provide a more comprehensive discussion of why contradictions with existing literature might exist and what they imply for the field.

10. Integration of Molecular and Clinical Perspectives: Integrate the molecular findings with clinical observations and implications for a more holistic understanding of the topic.

Response: We thank the reviewer for all the constructive points and comments. Accordingly, the Discussion has been extensively re-written (pp. 11-17). As suggested, in the updated text, among other topics, we emphasized the comparison of our results to the relevant literature, discussed reasons for the contradicting findings, raised issues for future investigations and noted study limitations. In regard to the impact of our results on public health, we may say that our contribution lies in the fact that we did not find a higher risk in patients with oral lichen planus or oral lesions with dysplasia for getting infected with SARS-CoV-2. Back during the days of the beginning of the pandemia, these patients kept asking us with much anxiety about their risk and then we did not have any answer. We summarized this on p. 17, lines 727-736. 

Reviewer #3: The reviewer appreciates the submission and has a few suggestions to improve the manuscript.

1) The lack of ACE2 receptors in squamous epithelium and the inconsistent furin expression has already been published, but the role of macrophages remains unclear and needs to be discussed. The detection of macrophages is somehow laborious in the given biopsies, but it should be added.

Response: We thank the reviewer for this interesting point. We understand that the paper by Bräutigam et al (Histopathology. 2023 May;82(6):846-859), as the Reviewer noted in point #4, is the base for this suggestion. We would like to comment that we read carefully the mentioned publication. In regard to the expression of ACE2 in alveolar macrophages (AM) the median expression was 0 (mean 0.6) (Table 3), which is rather similar to our results. In general, identification of proteins in macrophages might be the result of ingestion of cell debris and not necessarily inherent genetic expression of these proteins by macrophages themselves. Expression of furin was found in AM (median 2, mean 2.2), and we also found furin expression in the oral mucosa, however since the main SARS-CoV-2 entry receptor is ACE2, its expression on target cells is the critical issue. In addition, there is a remarkable morphological difference 

between the oral and the alveolar types of epithelia, which might also be reflected in local adaptations of the AM, including expression of selected factors that are important for the functionality of the alveolar epithelium. Therefore, investigation of the macrophages for expression of ACE2 and furin in the oral mucosa, might not be as relevant as in the lung alveoli, but could definitely be the focal point of a further study. 

2) It would be of great interest to stain mucosal samples from patients with SARS Cov 2 Infection, maybe ssome can be obtained from autoptic material.

Response: It was not the aim of our study to investigate oral tissues of infected patients, but rather the baseline of expression of ACE2 and related entry host factors in order to assess the vulnerability of these tissues to be infected by SARS-CoV-2. 

3) The figures with the immunohistochemical stainings need to be improved.

Response: Quality of photomicrographs was improved.

4) Literature is missing: only two examples.....

Determinants of SARS-CoV-2 entry and replication in airway mucosal tissue and susceptibility in smokers.

Nakayama T et al. Cell Rep Med. 2021 Oct 19;2(10):100421. doi: 10.1016/j.xcrm.2021.100421. Epub 2021 Sep 28.

PMID: 34604819

Comprehensive analysis of SARS-CoV-2 receptor proteins in human respiratory tissues identifies alveolar macrophages as potential virus entry site.

Bräutigam K, Reinhard S, Wartenberg M, Forster S, Greif K, Granai M, Bösmüller H, Klingel K, Schürch CM.

Histopathology. 2023 May;82(6):846-859. doi: 10.1111/his.14871. Epub 2023 Feb 21.

PMID: 36700825

Response: The paper by Nakayama et al was added and it definitely added valuable points to support our findings (current reference #16). In general, our reference list has been expanded according to changes done throughout the manuscript.

The reference in regard to alveolar macrophages was addressed in reviewer's point #1. 

6. PLOS authors have the option to publish the peer review history of their article (what does this mean?). If published, this will include your full peer review and any attached files.

Do you want your identity to be public for this peer review? For information about this choice, including consent withdrawal, please see our Privacy Policy.

Reviewer #1: No

Reviewer #2: No

Reviewer #3: No

---

## [Decision Letter · Decision Letter 1]

26 Feb 2024

Pathological changes in oral epithelium and the expression of SARS-CoV-2 entry receptors, ACE2 and furin

PONE-D-23-28297R1

Dear Dr. Vered,

We’re pleased to inform you that your manuscript has been judged scientifically suitable for publication and will be formally accepted for publication once it meets all outstanding technical requirements.

Kind regards,

Cheorl-Ho Kim, Ph.D.

Academic Editor

PLOS ONE

Additional Editor Comments (optional):

Dear Dr Vered

Thank you for your revision and appropriate responses.

I have long been considered your revision and found than that the revision has been appropriately made, as judged by the revision response.

Several issues raised by independent experts are well addressed.

Thank you again

Sincerely

Cheorl-Ho Kim

Editor

Reviewers' comments:

Reviewer's Responses to Questions

**Comments to the Author**

1. If the authors have adequately addressed your comments raised in a previous round of review and you feel that this manuscript is now acceptable for publication, you may indicate that here to bypass the “Comments to the Author” section, enter your conflict of interest statement in the “Confidential to Editor” section, and submit your "Accept" recommendation.

Reviewer #2: All comments have been addressed

2. Is the manuscript technically sound, and do the data support the conclusions?

Reviewer #2: Yes

3. Has the statistical analysis been performed appropriately and rigorously? 

Reviewer #2: (No Response)

4. Have the authors made all data underlying the findings in their manuscript fully available?

Reviewer #2: (No Response)

5. Is the manuscript presented in an intelligible fashion and written in standard English?

Reviewer #2: Yes

6. Review Comments to the Author

Reviewer #2: In reviewing the manuscript, I have carefully considered the responses provided to the questions posed by the journal. I am satisfied to note that all of my comments and concerns have been thoroughly addressed by the authors. Their efforts in revising the manuscript and providing detailed explanations and clarifications where necessary are commendable. The authors have demonstrated a commitment to research integrity, ensuring that issues related to dual publication, research ethics, and publication ethics have been carefully considered and appropriately addressed. This manuscript now significantly contributes to the field, and I have no further concerns or suggestions at this point. I believe the manuscript is now ready for publication, offering valuable insights and advancements to its readership.

7. PLOS authors have the option to publish the peer review history of their article (what does this mean?). If published, this will include your full peer review and any attached files.

Reviewer #2: No

---

## [Editor Report · Acceptance letter]

7 Mar 2024

PONE-D-23-28297R1 

PLOS ONE

Dear Dr. Vered, 

I'm pleased to inform you that your manuscript has been deemed suitable for publication in PLOS ONE. Congratulations! Your manuscript is now being handed over to our production team.

Kind regards, 

on behalf of

Professor Cheorl-Ho Kim 

Academic Editor

PLOS ONE